# A New *Aculodes* Species (Prostigmata: Eriophyidae) Described from an Invasive Weed by Morphological, Morphometric and DNA Barcode Analyses [note 1]

**DOI:** 10.3390/insects13100877

**Published:** 2022-09-27

**Authors:** Biljana Vidović, Nikola Anđelković, Vida Jojić, Tatjana Cvrković, Radmila Petanović, Francesca Marini, Massimo Cristofaro, Brian G. Rector

**Affiliations:** 1Department of Entomology and Agricultural Zoology, Faculty of Agriculture, University of Belgrade, Nemanjina 6, 11080 Belgrade, Serbia; 2Department of Genetic Research, Institute for Biological Research “Siniša Stanković”—National Institute of Republic of Serbia, University of Belgrade, Bulevar Despota Stefana 142, 11060 Belgrade, Serbia; 3Institute for Plant Protection and Environment, Banatska 33, 11080 Belgrade, Serbia; 4Serbian Academy of Sciences and Arts, Knez Mihailova 35, 11000 Belgrade, Serbia; 5Biotechnology and Biological Control Agency (BBCA), Via Angelo Signorelli 105, 00123 Rome, Italy; 6USDA-ARS, Great Basin Rangelands Research Unit, 920 Valley Road, Reno, NV 89512, USA

**Keywords:** Acari, biological control of weeds, CVA, Eriophyoidea, MANOVA, Poaceae

## Abstract

**Simple Summary:**

Natural enemies of cheatgrass, an invasive plant in the western USA that is associated with recent increases in destructive wildfires, were sought as part of a classical biological control program targeting that plant. A population of mites was discovered infesting cheatgrass plants in central Bulgaria and determined to be a new species using morphological, morphometric analysis (i.e., measurement of specific body parts) and DNA barcoding (i.e., comparison of DNA sequences in genes shared by all mite species). Without these modern analytical tools, this mite would probably have been considered as the same species as another mite that infests other plants. This new mite species will now be tested to determine if it is suitable for importation and release in the USA as a biological control agent of cheatgrass.

**Abstract:**

A new species of eriophyoid mite, *Aculodes marcelli*
**sp. nov.**, was discovered on cheatgrass, *Anisantha tectorum* (L.) Nevski (syn. *Bromus tectorum* L.), an annual grass that is native to Eurasia and Northern Africa. This grass was introduced to North America near the end of the 19th century and now is widespread and associated with the observed increases in the size, frequency, and intensity of wildfires in western N. America. In this paper, *A. marcelli*
**sp. nov.**, is morphologically described and illustrated. Compared with other *Aculodes* spp., it differs based on morphology and the sequence of the mitochondrial cytochrome oxidase gene, subunit I (MT-CO1). Results of morphometric analysis showed clear differentiation between *A. marcelli*
**sp. nov**., and the most similar congener, *A. altamurgiensis* from *Taeniatherum caput-medusae*. Analysis of MT-CO1 sequence divergence revealed significant levels of genetic variation (17.7%) and supported the results from the morphometric analysis; therefore, it is determined that they are two different species. *Aculodes marcelli*
**sp. nov.**, is a new candidate agent for classical biological control of *A. tectorum*.

## 1. Introduction

Eriophyoid mites are obligate herbivores characterized by minute size and simplified body structure [1]. Traditional descriptions of new eriophyoid species only on the basis on morphological traits often provide inadequate separation between taxa, especially in cases where species are morphologically very similar. Most contemporary reports concerning alpha-taxonomy are prepared following traditional methods, but there are more and more examples of new species descriptions of eriophyoids, or splitting from the putative complex of taxa, using methods of traditional taxonomy combined with the support of DNA sequences of one (usually COI mitochondrial gene) or a few gene fragments. The potential for high host specialization, host-race formation, and cryptic speciation is particularly great in eriophyoid mites because they are often intimately associated with their hosts, lack long-range host seeking ability, and have high reproductive rates [2,3]. A contemporary approach in describing species diversity, so called integrative taxonomy, has recently revealed the occurrence of cryptic species in several eriophyoid mite genera [4,5,6,7]. This integrative approach to the taxonomy of Eriophyoidea has gained traction with contemporary authors and in many recent studies traditional descriptions are integrated with morphometric analyses, molecular tools, and other characteristics [8,9,10,11]. 

Searching for biological control agents of cheatgrass, *Anisantha tectorum* (L.) Nevski (syn. *Bromus tectorum* L.), an eriophyid mite belonging to the genus *Aculodes,* was found that was morphologically similar to *Aculodes altamurgiensis* de Lillo & Vidović, a recently described species from medusahead (*Taeniatherum caput-medusae* (L.) Nevski) [12]. In preliminary morphological investigations, some morphological differences were observed between the cheatgrass mite and *A. altamurgiensis* and detailed analyses were required to discriminate these morphologically similar mite entities. Moreover, comparative analysis of discriminatory characteristics of *Aculodes* spp., described to date was necessary to ensure that the mite from cheatgrass was a new species.

To date, about 30 species of eriophyid mites belonging to the genus *Aculodes* have been described worldwide. Most of them (approx. 26) have been recorded from grasses (Poaceae), while the others were found on hosts within the families Malvaceae, Salicaceae, Fabaceae, and Rosaceae [13,14,15,16,17,18,19,20,21,22,23,24,25,26,27,28,29,30,31,32,33,34,35,36,37,38,39,40,41]. Until now, only two species of the genus *Aculodes* (*A. multitricavus* and *A. janboczeki*) have been described from *Bromus* spp., both from *B. inermis* Leyss. (syn. of *Bromopsis inermis* (Leyss.) Holub) [30,31] and by preliminary research of Kiedrowicz et al. [42], one new species belonging to the genus *Aculodes* is registered on *Bromus armenus* Boiss. (syn. of *Bromopsis armena* (Boiss.) Holub). 

Cheatgrass is a typically winter annual grass that reproduces only by seed [43]. It is native in temperate regions throughout much of Eurasia and Northern Africa. The introduction of cheatgrass to North America probably occurred independently several times. It was first reported in the far western United States near the end of the 19th century and is now widespread throughout most of the U.S., Canada, Greenland, and northern Mexico [44,45,46].

The aim of this study is the description of the new eriophyid mite species, *Aculodes marcelli*
**sp. nov.,** from *Anisantha tectorum* using traditional morphological methods, morphometric analysis, and molecular tools.

## 2. Materials and Methods

### 2.1. Collection and Morphological Measurements

Due to the morphological similarity of the above mentioned *Aculodes* mites registered on cheatgrass and medusahead, mites collected from *Anisantha tectorum*, specimens of *A. altamurgiensis* from medusahead were also collected as comparative material.

Plant samples of *Anisantha tectorum* were collected in Starosel, Bulgaria (42.485°; 24.5522°) and Surduk, Serbia (45.0925°; 20.2975°); *Taeniatherum caput-medusae* plants were collected in Starosel, Bulgaria (42.485°; 24.5522°) and Andria, Italy (41.1172°; 16.4183°). 

Mite specimens from each plant sample were mounted in Keifer’s F medium [47] and then examined using a Leica DMLS research microscope with phase-contrast and the software package IM 1000 (Leica, Wetzlar, Germany).

Material for morphological study included specimens of *Aculodes altamurgiensis* from *T. caput-medusae* collected in Italy and *A. marcelli*
**sp. nov.,** from *A. tectorum* collected in Bulgaria and Serbia. For the morphometric analyses, 25–27 females from each sample were examined in the dorso-ventral position. Among the total of 23 morphological characters, two (number of dorsal and ventral rings) were meristic. All analyzed variables were distributed normally according to Kolmogorov–Smirnov tests. A one-way multivariate analysis of variance (MANOVA) was used to examine the differences in morphological variation among the three mite populations. Canonical variate analysis (CVA) was performed to visualize differences and to determine the relative importance of characters as discriminators among the studied mites. All statistical analyses were conducted using the Statistica 6 software package [48].

Material for species description included only the population of *A. marcelli*
**sp. nov.,** collected from *A. tectorum* in Bulgaria. For the species description of *A. marcelli*
**sp. nov.**, measurements are given in micrometers (µm) and, unless stated otherwise, refer to the length of the structure. Each measurement of the holotype precedes the corresponding range for paratypes (given in parentheses). The morphology and nomenclature follow [1] and genus classification is based on [49]. Measurements and illustrations were made according to Amrine & Manson [47] and de Lillo et al. [50]. 

Host plant names and their synonyms are in accordance with the on-line database: Euro+Med PlantBase [51]. The holotype and paratype slides as well as all material examined were deposited in the Acarology Collection, Department of Entomology and Agricultural Zoology, Faculty of Agriculture, University of Belgrade, Serbia.

### 2.2. Scanning Electronic Microscopy

Scanning electron micrographs (SEM) were made according to Alberti & Nuzzaci [52] using a scanning electron microscope (JEOL-JSM 6390, JEOL GmbH, Munich, Germany) at the Faculty of Agriculture, University of Belgrade. Live mites were collected individually with a fine entomological needle from fresh plant parts under a stereomicroscope (Leica M60 Leica, Wetzlar, Germany) and placed in the SEM sample holder. Prior to the observation, holder mounted mites were gold-coated (BAL-TEC SC-RD005, Balzers, Liechtenstein) by a Sputter coater (BAL-TEC AG, Balzers, Liechtenstein) for 100 s under a 30 mA ion current.

### 2.3. DNA Extraction, PCR Amplification and Sequencing

Material subjected to molecular analysis included populations inhabiting *A. tectorum* from Bulgaria and Serbia and the population associated with *T. caput-medusae* from Bulgaria. Mite specimens collected from fresh leaves were preserved in 96% ethanol and stored at 4 °C until DNA extraction. Total DNA was extracted from pools of one to three whole specimens using QIAGEN DNeasy Blood & Tissue Kit (QIAGEN, Hilden, Germany), according to the manufacturer’s instructions, with modifications based on the work of Dabert and colleagues [53]. The barcode region of the mitochondrial cytochrome oxidase (MT-CO1) gene was amplified using LCO1490 and HCOd primers [54,55]. The polymerase chain reactions (PCR) contained High Yield Reaction Buffer A with MgCl_2_ (1x), additional 2.5 mM MgCl_2_, 0.6 mM of each dNTP, 0.6 μM of each primer, 1U of FastGene Taq DNA polymerase (NIPPON Genetics Europe) and 5 μL of diluted template DNA in a 25 μL final volume. PCR was carried out in a Mastercycler ep gradient S thermal cycler (Eppendorf, Hamburg, Germany) following the protocol and amplification conditions of [56]. PCR amplicons were visualized in a 1% agarose gel and purified using the QIAquick PCR purification Kit (QIAGEN) according to the manufacturer’s instructions. Sequencing was performed in both directions with the same primer pairs as in the initial PCR procedure by Macrogen Europe (Amsterdam, Netherlands).

### 2.4. Alignments, Molecular Analyses and Genetic Distances

Both strands of each barcode MT-CO1 amplicon were assembled into contigs using FinchTV v.1.4.0 (www.geospiza.com; accessed on 21 September 2022), translated into amino acids to check for the absence of stop codons and aligned by CLUSTAL W integrated in MEGA11 software [57]. Sequences of *A. marcelli*
**sp. nov**., from *Anisantha tectorum* populations collected in Serbia and Bulgaria and *A. altamurgiensis* associated with *T. caput-medusae* collected from Bulgaria were trimmed and compared with corresponding sequences of *A. altamurgiensis* (GenBank accession numbers MH352403 and MH352404), *A. mckenziei* (Keifer) (FJ387561 and FJ387562), and *A. holcusi* Skoracka (MW439276). Uncorrected pairwise genetic distances were used to calculate the average genetic distance between the sequences.

## 3. Results

### 3.1. Morphometric Analysis

Means and standard deviations for the total of 23 morphological traits are given in Table 1. The one-way MANOVA for 23 morphological characters revealed statistically significant differences among the analyzed mite populations (Wilks’ Lambda = 0.0832, F_46, 104_ = 5.57, *p* < 0.0001). As illustrated by a CVA scatter plot of the first two canonical variates (CV) (Figure 1), the CV1 axis (accounting for 77.6% of the total morphological differences) separated mites from *T. caput-medusae* from those hosted by *A. tectorum* collected in Serbia, while a segregation of mites from *A. tectorum* collected in Bulgaria and those hosted by *A. tectorum* collected in Serbia is visible along the CV2 axis (accounting for 22.4% of the total variation). The total percentage of correctly classified mite specimens from each of the three plant samples was 94.8% (100% for mites hosted by *A. tectorum* collected in Serbia; 96.3% for mites hosted by *T. caput-medusae*; 88% for mites hosted by *A. tectorum* collected in Bulgaria).

The morphological characters that contributed most to the separation of the analyzed mite populations along the CV1 axis were the lengths of setae *sc*, *d*, and *c2*, distance between *1a* tubercles, and the number of dorsal annuli (Table 2). In comparison to the mites from *T. caput-medusae* (negative section of the CV1 axis in Figure 1), those hosted by *A. tectorum* collected in Serbia (positive section of the CV1 axis in Figure 1) were characterized by higher mean values for these characters (Table 1). Morphological characters that contributed most to the segregation of the analyzed mite populations along the CV2 axis were the lengths of setae *sc*, *c2*, and *3a*, distance between 2*a* tubercles, length of tibia II, and width of genitalia. In comparison to the mites from *A. tectorum* collected in Bulgaria (negative section of the CV2 axis in Figure 1), those hosted by *A. tectorum* collected in Serbia (positive section of the CV2 axis in Figure 1) showed higher mean values for these characters, except for the distance between 2*a* tubercles and the length of tibia II (Table 1).

### 3.2. Molecular Analyses

A mitochondrial CO1 fragment of 606 bp in length was generated from all *Aculodes* specimens. Nucleotide sequence data were deposited in the GenBank database under the accession numbers indicated in Table 3. The translation of the nucleotide sequences resulted in 202 amino acid positions. Sequence comparison showed that populations of *A. marcelli*
**sp. nov.**, collected on *A. tectorum* in Bulgaria and Serbia, are 100% identical. 

Pairwise comparison of trimmed MT-CO1 fragments (531 bp) between *A. marcelli*
**sp. nov.,** and *A. altamurgiensis* averaged 17.6% genetic divergence. Similar p-distances were estimated between the new species and *A. holcusi* (17.2%), while average pairwise distance estimated between *A. marcelli*
**sp. nov.,** and *A. mckenziei* was 21.8% (Table 3).

Translation into amino acid sequences revealed 1.1% divergence between populations of *A. marcelli*
**sp. nov.,** with both *A. altamurgiensis* and *A. holcusi*, which is approximately one-quarter the amino acid divergence between the new species and *A. mckenziei* (4%) (Table 3).

### 3.3. Taxonomy

***Aculodes marcelli*** Vidović **sp. nov**., (Figure 2, Figure 3 and Figure 4)

**Description.** FEMALE (n = 10). Body wormlike 277 (217–293), 60 (57–65) wide, whitish in color. **Gnathosoma** 19 (18–20) curved downwards, chelicerae 16 (14–16), setae *ep* 3, setae *d* 10 (8–10) unbranched. **Prodorsal shield** 34 (34–37) including the frontal lobe, 38 (33–38) wide; triangular with a pointed frontal lobe over the gnathosoma; median line present on rear half of the shield; admedian lines complete, from anterior lobe diverging to lateral margin of shield; I pair of submedian lines present on rear half, incomplete, parallel to admedian; II pair of submedian lines present, incomplete, arched. I and II submedian lines connecting in the 1/4 anterior part of the shield; conical microtubercles and dashes present between the lines on rear surface of the shield. Tubercles *sc* on rear shield margin 16 (16–20) apart, scapular setae *sc* 39 (38–54). **Leg I** 38 (37–42); femur 12 (9–12), setae *bv* 13 (13–15); genu 6 (5–7), setae *l*″ 22 (22–27); tibia 8 (8–9), setae *l′* 9 (9–13); tarsus 8 (8–10), setae *ft′* 17 (17– 21), setae *ft″* 27 (26–28); tarsal solenidion *ω* 8 (8–10) with a thinner and rounded end; tarsal empodium 9 (8–10), 8-rayed. **Leg II** 34 (34–38); femur 12 (10–12), setae *bv* 18 (18–26); genu 6 (5–6), setae *l″* 14 (13–16); tibia 7 (6–7); tarsus 9 (8–9), setae *ft′* 9 (9–11), setae *ft″* 26 (24–28); tarsal solenidion *ω* 8 (8–10) similar to that on leg I; tarsal empodium 9 (8–10), 8-rayed. **Coxae** with a pattern of wavy lines and dashes; sternal line 11 (11–13); setae *1b* 9 (9–11), tubercles *1b* 10 (9–11) apart; setae *1a* 21 (19–28), tubercles *1a* 7 (6–9) apart, setae *2a* 43 (27–44), tubercles *2a* 23 (21–25) apart. **Genital coverflap** 12 (11–15), 23 (20–23) wide, with 9 (9–12) longitudinal striae in a single row; setae *3a* 19 (18–23), 17 (14–17) apart. **Internal genitalia** with anterior apodeme trapezoidal, longitudinal bridge relatively long, the post-spermathecal part of the longitudinal bridge is reduced; spermathecal tubes directed latero-posterad, composed of two parts: basal part egg-shaped, distal more tubulose; spermathecae globose. **Opisthosoma** with subequal annuli: 63 (63–69) dorsal and 66 (61–68) ventral annuli; 5 (5–6) coxigenital annuli. Dorsal and ventral opisthosoma with pointed microtubercles close to the rear margins of annuli. Setae *c2* 29 (29–40), 51 (46–56) apart, on annulus 9 (8–10); setae *d* 38 (34–49), 35 (30–38) apart, on annulus 21 (21–23); setae *e* 34 (31–35), 16 (13–18) apart, on annulus 37 (35–41); setae *f* 24 (24–27), 18 (16–19) apart, on annulus 60 (57–62); seta *h2* 66 (55–79), 9 (7–9) apart; setae *h1* 7 (6–9), 5 (4–5) apart.

MALE (n = 2). Body wormlike, 197–200, 53–54 wide, whitish in color. **Gnathosoma** 17 curved down, cheliceral stylets 14–15. **Prodorsal shield** 33, 33–34 wide. Prodorsal shield tubercles on the rear shield margin 18–19 apart, setae *sc* 34–39, projecting posteriorly. Shield design similar to female. **Leg I** 30–33; femur 9, setae *bv* 9–10; genu 5, setae *l″* 21–24, tibia 6–7, tibial setae *l′* 10; tarsus 8, setae *ft′* 16, setae *ft″* 21–23; solenidion *ω* 8, empodium *em* 6, and 8-rayed. **Leg II** 28–29; femur 8–9, setae *bv* 17–18; genu 4–5, genual setae *l″* 13–15, tibia 5; tarsus 7–8; setae *ft′* 8–11, setae *ft″* 21–22; solenidion *ω* 9–10, empodium *em* 6–7. Coxae granulated; sternal line 10–11; setae *1b* 10–12, *1b* tubercles 9–10 apart; setae *1a* 22–27, *1a* tubercles 5 apart; setae *2a* 29–34, *2a* tubercles 18 apart. **Genitalia** 19–20 wide; setae *3a* 20, *3a* tubercles 14–15 apart. **Opisthosoma** with subequal annuli: 61–66 dorsal and 61–63 ventral annuli; 4–5 coxigenital annuli. Setae *c2* 35–37, 41–44 apart, on annulus 8–10; setae *d* 42–44, 31–33 apart, on annulus 19–21; setae *e* 26–30, 15–16 apart, on annulus 36; setae *f* 16–24, 18 apart, on annulus 57–59; setae *h2* 50, 7–8 apart, setae *h1* 5, 4 apart.

**Type host plant.***Anisantha tectorum* (L.) Nevski (Poaceae), cheatgrass.

**Type locality.** Starosel, Bulgaria 42.485°; 24.5522°, 396 m elev.

**Type material.** Female holotype (slides # 862/1) and paratypes 42 females, 4 males; 3 May 2018, collected by Brian Rector. 

**Additional studied material.** Starosel, Bulgaria (42.485°; 24.5522°), 8 May 2017, 12 slides; 20 May 2018, 10 slides collected by F. Di Cristina, F. Marini, and M. Cristofaro. Surduk, Serbia (45.0925°; 20.2975°), 15 May 2018, 46 slides, collected by D. Smiljanić and B. Vidović.

**Relation to the host.** The mites are vagrants on the leaf surfaces and in spikes, sometimes in large colonies. No visible damage was observed although effects on the host’s life cycle have yet to be tested.

**Etymology.** The species name, *marcelli,* is derived from the name of Marcello Barlattani, a member of our technical support crew who died during the course of this work, in honor and memory of him.

**Differential diagnosis and remarks.** All *Aculodes* spp., from Poaceae can split out in the three groups based on the presence and length of the median line on the prodorsal shield and based on the number of empodial rays (Table 4). *Aculodes marcelli*
**sp. nov**., has an incomplete median line in the posterior half of the shield, which is different from the species that lack a median line (*A. capillarisi* Skoracka, *A. dubius* (Nalepa), *A. janboczeki* Skoracka, *A. koeleriae* Sukhareva, *A. kransnovi* Sukhareva, *A. stoloniferae* Skoracka, *A. sylvatici* Skoracka, Labrzycka & Rector) and species with a complete median line (*A. bambusae* Kuang, *A. levis* Huang, *A. ponticus* Sukhareva).

*Aculodes marcelli***sp. nov.**, has an incomplete median line in the posterior half of the shield, which is different from the species that lack a median line (*A. capillarisi* Skoracka, *A. dubius* (Nalepa), *A. janboczeki* Skoracka, *A. koeleriae* Sukhareva, *A. kransnovi* Sukhareva, *A. stoloniferae* Skoracka, *A. sylvatici* Skoracka, Labrzycka & Rector) and species with a complete median line (*A. bambusae* Kuang, *A. levis* Huang, *A. ponticus* Sukhareva). 

*Aculodes marcelli***sp. nov.**, can be distinguished from other *Aculodes* spp., that also have an incomplete median line on the basis of the number of empoidal rays (*Aculodes marcelli* sp. nov., 8-rayed; *A. multitricavus* Skoracka, *A. mckenziei* (Keifer), and *A. stipacolus* Alemandri & Navia 9-rayed; *A. altamurgiensis* de Lillo & Vidović, *A. festucae* Skoracka, Labrzycka & Rector, *A. fuller* (Keifer), *A. tsukushiensis* Xue, Song & Hong, *A. caespiticolus* Ripka & Szabo, *A. tinaniensis* Elhalawany & Amrine, and *A. skorackae* Elhalawany & Amrine 7-rayed).

Among the *Aculodes* spp., with incomplete median lines and eight empodial rays, *Aculodes marcelli* sp. nov., may differ on the basis on presence and position of their submedian lines, the presence of microtubercles on the prodorsal shield and the shape of the microtubercles on the dorsal and ventral side of the opisthosoma.

Differential characteristics of *Aculodes marcelli*
**sp. nov.,** in comparison with the most similar species are the following: dashes are present between the lines of prodorsal shield, two submedian lines which touch each other at their anterior part are present, of which the second is longer. *A. agropyronis* differs by the absence of dashes between the lines, *A. calamaabditus* and *A. deschampisae* differ by the presence of only I submedian line, while *A. mongolicus* has a longer I submedian line, and in *A. neglectivagrans* submedian lines I and II do not touch each other. The dorsal and ventral opisthosoma in *Aculodes marcelli*
**sp. nov.,** have pointed microtubercles, unlike the *A. holcusi* opisthosoma, which has subrounded dorsal and slightly pointed ventral microtubercles.

With regard to the prodorsal shield design *Aculodes marcelli* sp. nov., is most similar to *A. altamurgiensis*, in that they have the same design of prodorsal shield. The only obvious characteristics that distinguish these two species are the number of empodial rays (*A. marcelli* sp. nov., 8-rayed; *A. altamurgiensis,* 7-rayed) and the number of dorsal (*A. marcelli* sp. nov., 63–69; *A. altamurgiensis,* 51–58) and ventral annuli (*A. marcelli* sp. nov., 61–68; *A. altamurgiensis,* 55–66).

## 4. Discussion

Variability of phenotypic traits among populations of mites inhabiting different host plants may occur as different mite species (i.e., total separation of gene pools); host races (i.e., partial differentiation of gene pools); or intraspecific phenotypic plasticity (i.e., no separation of the gene pool) [58]. As a result of these multiple possible causes of phenotypic variability, taxonomic assessment based on traditional, strictly morphological methods may fail to adequately separate species, especially in the case of cryptic species. Morphometric analysis may indicate that there is variability but without molecular analysis, it is impossible to declare two morphologically different eriophyoid populations collected from different hosts to be different species, short of elaborate and time-consuming cross-breeding and host-preference bioassays [4,6,8,59,60,61]. Molecular tools and methods, including sequencing and publishing barcode genes in publicly available databases, are useful in discrimination of cryptic species and should be included with alpha taxonomy whenever it is possible [4,5,7,9,62,63,64].

Analysis of MT-CO1 sequence divergence in the present study supported the morphometric analyses, revealing significant levels of genetic variation and host-associated genetic structure between *Aculodes* spp., mites collected on *A. tectorum* and on *T. caput-medusae.* Although *A. marcelli*
**sp. nov**., is morphologically very similar to *A. altamurgiensis*, interspecific nucleotide sequence divergence of MT-CO1 was 17.7%, which corresponds to p-distance values between MT-CO1 sequences available in databases of other morphologically different congeneric mite species that are also associated with closely related host plants (e.g., *A. holcusi* and *A. mckenziei*). Similarly, clear differentiation was shown in other eriophyoid genera inhabiting different grass hosts such as *Aceria tosichella* s.l., 13.9% [8] and *Abacarus hystrix,* 22.6% [65].

Because of the great morphological similarity between already described *Aculodes altamurgiensis* and recently discovered *Aculodes marcelli*
**sp. nov.**, there was an early suspicion that they were the same species, but from different host plants (BV, pers. comm.). However, the integrative approach for description of the new species of eriophyoid mites presented herein, i.e., results of both phenotypic and genotypic variability did not confirm this hypothesis. According to the results of our morphometric analysis, *A. altamurgiensis* is clearly separated from two populations of *A. marcelli*
**sp. nov.**, and the analysis of MT-CO1 sequence divergence revealed significant levels of genetic variation (17.7%), indicating that they are indeed two different species. However, while some morphometric differences were observed between two populations of *A. marcelli*
**sp. nov.**, collected from Bulgaria and Serbia, no genetic variability was identified between them, which leads us to the conclusion that it is a single species exhibiting phenotypic plasticity in two reproductively isolated populations.

Like *T. caput-medusae*, *A. tectorum* is an important weed in western N. America [66]. The dried, senesced plants of both species persist in arid environments and provide highly flammable tinder to ignite and carry fire to larger fuel sources; this contributes significantly to the recent increases in the frequency, size, and intensity of wildfires in that region [67,68]. The United States Department of Agriculture, Agricultural Research Service leads an international classical biological control program targeting both *A. tectorum* and *T. caput-medusae*. In order to determine its suitability as a classical biological control agent of *A*. *tectorum*, *Aculodes marcelli*
**sp. nov.**, will undergo similar biological testing as is being performed to evaluate *A. altamurgiensis* as a candidate agent for control of *T. caput-medusae* [69]. The fact that *A. marcelli*
**sp. nov.**, was never described prior to the discovery reported herein provides a strong indication that it does not cause damage to cereal species or other economically important plants.

## Figures and Tables

**Figure 1 insects-13-00877-f001:**
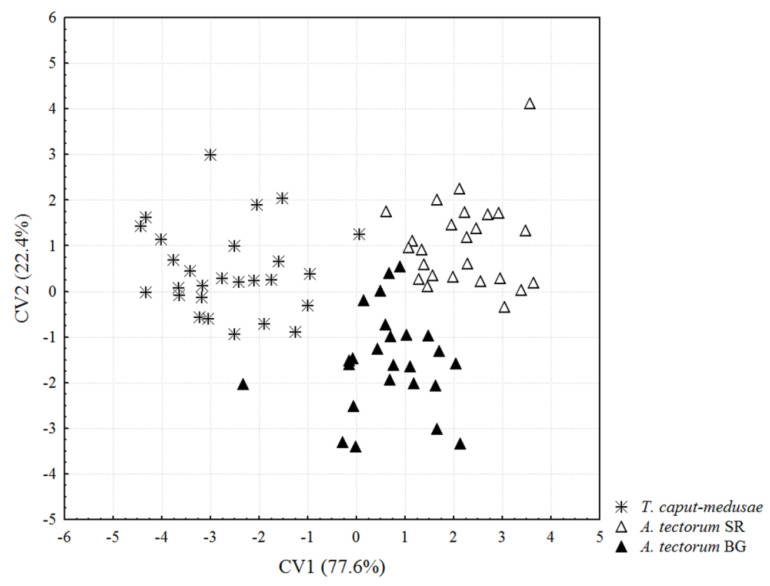
Canonical variate analysis (CVA) of 23 morphological traits. “*T*. *caput-medusae”* = *Aculodes altamurgiensis* from *Taeniatherum caput-medusae* collected in Italy, “*A. tectorum* SR” = *Aculodes marcelli*
**sp. nov.,** from *Anisantha tectorum* collected in Serbia, “*A. tectorum* BG” = *Aculodes marcelli*
**sp. nov.,** from *A. tectorum* collected in Bulgaria.

**Figure 2 insects-13-00877-f002:**
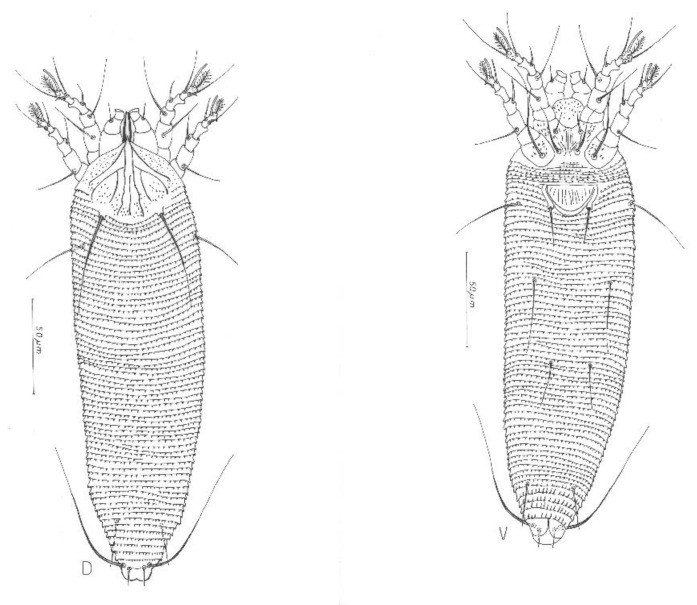
Schematic drawings of female *Aculodes marcelli*
**sp. nov.**: D=dorsal mite; V=ventral mite.

**Figure 3 insects-13-00877-f003:**
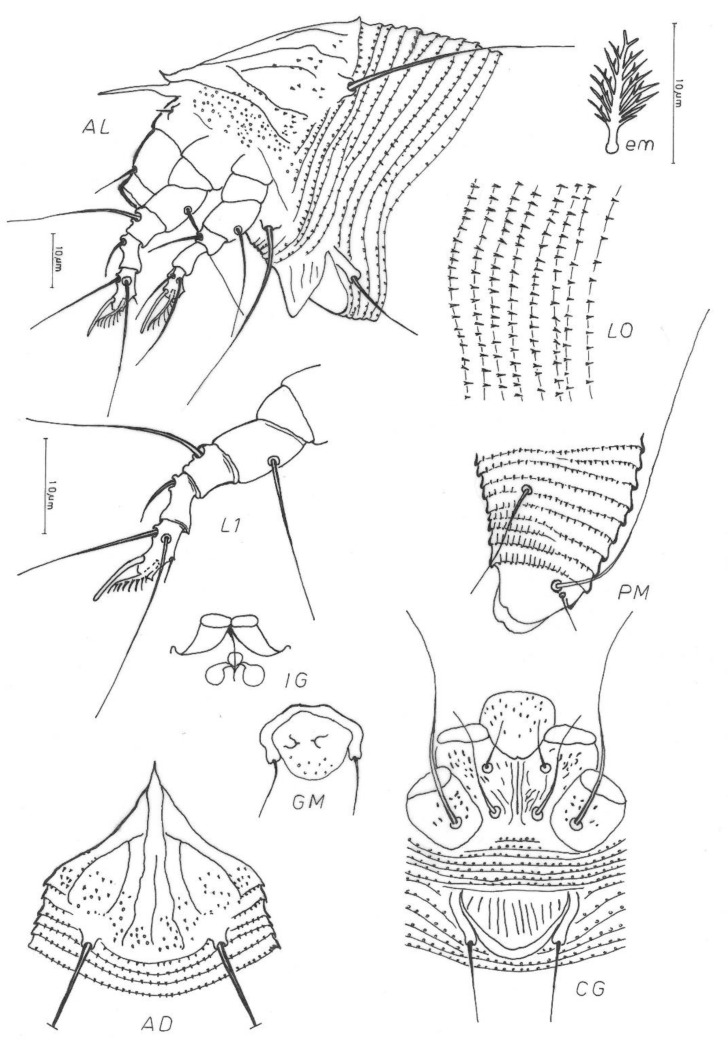
Schematic drawings of female *Aculodes marcelli*
**sp. nov.**: AD=Prodorsal shield; AL=Lateral view of anterior body region; CG=Female coxigenital region; em=Empodium; GM=Male genital region; IG=Internal female genitalia; LO=Lateral view of annuli; L1=Leg I; PM=Lateral view of posterior opisthosoma.

**Figure 4 insects-13-00877-f004:**
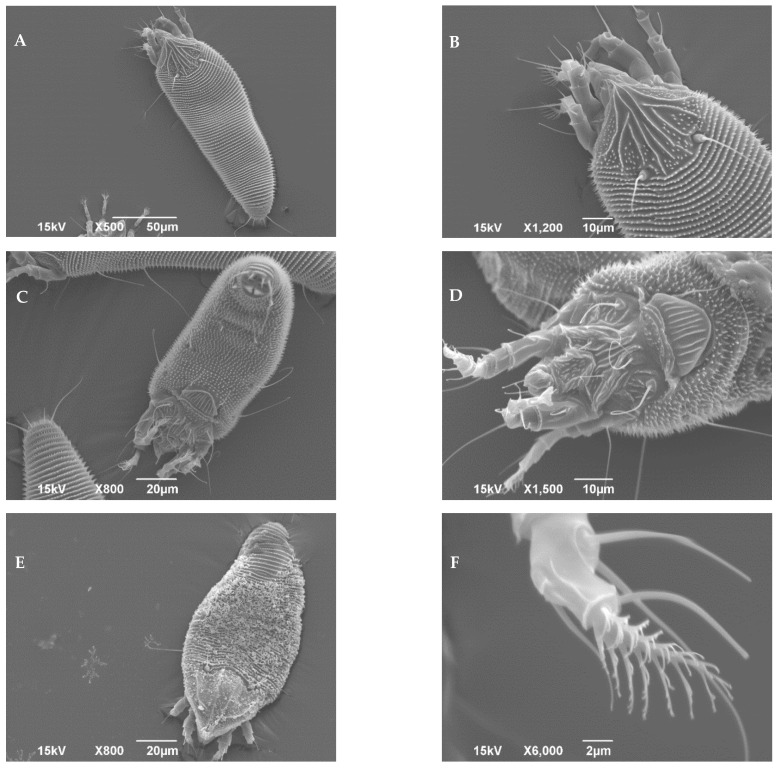
Scanning electron micrographs of ***Aculodes marcelli* sp. nov.**: (**A**) Dorsal view; (**B**) Dorsal view with detail of prodorsal shield; (**C**) Ventral view; (**D**) Ventral view, anterior half; (**E**) Dorsal view with apparent wax-like secretion; (**F**) Detail of tarsus.

**Table 1 insects-13-00877-t001:** Means and standard deviations (SD) for 23 morphological traits of *Aculodes altamurgiensis* from *Taeniatherum caput-medusae* collected in Italy and *Aculodes marcelli*
**sp. nov.,** from *Anisantha tectorum* collected in Bulgaria and Serbia (n = number of specimens).

Trait	*A. altamurgiensis* Italy (n = 27)		*A. marcelli* sp. nov. (n = 25) Bulgaria		*A. marcelli* sp. nov. (n = 25) Serbia	
	Mean	SD	Mean	SD	Mean	SD
A: body length	235.41	24.04	257.67	26.70	259.46	18.50
B: prodorsal shield length	35.68	1.59	36.67	1.48	37.49	1.58
C: prodorsal shield width	35.19	1.96	36.76	3.07	37.62	2.51
D: setae sc length	46.27	3.38	44.20	5.44	48.21	3.87
E: tubercles *sc* apart	18.38	1.03	18.83	1.37	19.25	1.14
F: number of dorsal annuli	61.81	2.77	66.96	2.94	68.96	3.69
G: number of ventral annuli	62.00	3.15	65.68	3.46	69.28	3.99
H: setae *c2* length	32.39	2.92	34.67	3.96	36.23	3.86
I: setae *d* length	37.88	3.57	42.98	4.68	47.96	4.47
J: setae *e* length	30.13	4.23	33.61	4.88	37.42	5.07
K: setae *f* length	25.01	2.24	26.27	1.80	28.52	2.56
L: genitalia length	11.19	1.05	11.53	0.99	11.80	1.49
M: genitalia width	20.22	0.75	20.29	1.18	20.93	0.83
N: setae *3a* length	19.77	2.06	22.01	3.29	22.03	2.81
O: tubercles *3a* apart	14.79	0.79	15.25	1.23	15.12	0.61
P: tubercles *1b* apart	8.34	0.77	9.67	1.02	9.66	0.95
Q: tubercles *1a* apart	5.71	0.64	6.25	0.96	6.04	0.68
R: tubercles *2a* apart	19.37	1.60	21.64	1.74	21.16	1.10
S: setae *2a* length	35.65	8.24	40.57	5.20	40.55	6.02
T: tibia I length	7.99	0.57	8.12	0.71	8.03	0.93
U: tarsus I length	8.06	0.62	8.39	0.52	8.91	0.61
V: tibia II length	6.56	0.56	6.81	0.55	6.69	0.83
W: tarsus II length	7.81	0.57	8.11	0.69	8.43	0.75

**Table 2 insects-13-00877-t002:** Standardized coefficients for canonical variables on the first two (CV1 and CV2) canonical axes (Cum. Prop. = cumulative proportion of described differences between mite populations. Morphological characters that contribute most to the separation of mite populations are marked in bold).

Trait	CV1	CV2
A: body length	0.066	−0.280
B: prodorsal shield length	−0.122	−0.047
C: prodorsal shield width	0.316	0.127
D: seta *sc* length	**−0.738**	**0.927**
E: tubercles *sc* apart	0.238	−0.325
F: no. of dorsal annuli	**0.524**	−0.253
G: no. of ventral annuli	−0.000	0.287
H: setae *c2* length	**0.474**	**−0.624**
I: setae *d* length	**0.556**	0.427
J: setae *e* length	0.121	0.105
K: setae *f* length	0.174	0.384
L: genitalia length	0.013	0.245
M: genitalia width	0.049	**0.583**
N: setae *3a* length	0.076	**−0.514**
O: tubercles *3a* apart	−0.092	−0.028
P: tubercles *1b* apart	0.350	−0.240
Q: tubercles *1a* apart	**−0.550**	0.470
R: tubercles *2a* apart	0.147	**−0.629**
S: setae *2a* length	−0.129	−0.429
T: tibia I length	−0.058	0.059
U: tarsus I length	0.407	0.401
V: tibia II length	0.084	**−0.505**
W: tarsus II length	0.303	0.098
Eigenvalue	4.336	1.251
Cum. Prop.	0.776	1.000

**Table 3 insects-13-00877-t003:** Pairwise distance matrix with uncorrected *p*-distances between *Aculodes marcelli*
**sp. nov**., and three congeners (*A. altamurgiensis, A. mckenziei,* and *A. holcusi*) based on CO1 nucleotide (below the diagonal) and amino acid sequences (above the diagonal). Numbers in column headers refer to species and populations listed in their respective rows. Genbank accession numbers provided in parentheses.

		1	2	3	4	5	6	7	8
**1**	*Aculodes marcelli***sp. nov.** Bulgaria (OP028122)		0.000	0.011	0.011	0.011	0.040	0.040	0.011
**2**	*Aculodes marcelli***sp. nov.** Serbia (OP028123)	0.000		0.011	0.011	0.011	0.040	0.040	0.011
**3**	*Aculodes altamurgiensis* Bulgaria (OP028124)	0.177	0.177		0.000	0.000	0.045	0.045	0.000
**4**	*Aculodes altamurgiensis* (MH352403)	0.177	0.177	0.032		0.000	0.045	0.045	0.000
**5**	*Aculodes altamurgiensis* (MH352404)	0.175	0.175	0.021	0.011		0.045	0.045	0.000
**6**	*Aculodes mckenziei* (FJ387561)	0.217	0.217	0.211	0.215	0.213		0.000	0.045
**7**	*Aculodes mckenziei* (FJ387562)	0.219	0.219	0.200	0.202	0.202	0.045		0.045
**8**	*Aculodes holcusi* (MW439276)	0.172	0.172	0.158	0.162	0.158	0.206	0.208	

**Table 4 insects-13-00877-t004:** Design of prodorsal shield and number of empodial rays of species in the genus *Aculodes* from grasses.

Without a Median Line		Complete Median Line	
*A. capillarisi*		*A. bambusae*	
*A. dubius*		*A. levis*	
*A. janboczeki*		*A. ponticus*	
*A. koeleriae*			
*A. krasnovi*			
*A. stoliniferae*			
*A. sylvatici*			
**Incomplete median line**			
7 rayed empodium	7–8 rayed empodium	8 rayed empodium	9 rayed empodium
*A. festucae*	*A. calamaabditus*	*A. agropyronis*	*A. mckenziei*
*A. fuller*	*A. deschampisae*	*A. holcusi*	*A. multitricavus*
*A. tsukushiensis*	*A. mongolicus*	*Aculodes marcelli* **sp. nov.**	*A. stipacolus*
*A. altamurgiensis*	*A. neglectivagrans*		
*A.caespiticolus*			
*A. tinaniensis*			
*A. skorackae*			

## Data Availability

All data produced in this study are reported herein.

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
