# Peer review of "A New Aculodes Species (Prostigmata: Eriophyidae) Described from an Invasive Weed by Morphological, Morphometric and DNA Barcode Analyses†"

_insects, 2022, doi:10.3390/insects13100877_

Round 1

Reviewer 1 Report

This work is very interesting and very good. It seems to me that it is better not to confuse morphology and DNA in one manuscript.  Because when we describe a new species, we concentrate as much as possible to describe and draw the new species. Here we don't see the complete drawing of the new species - the dorsal side, the ventral side of the idiosoma. And this is very important when we describe a new species. Separate parts of the idiosoma (like Figure 2, 243 row) we draw when the species is already known and when we do a re-description of the species. Please draw all mite - dorsal and ventral sides. Add - Starosel, Plovdiv region, Bulgaria (42.485Ëš; 24.5522Ëš).  In Starosel we have a mountain with a height of 1500 meters. Surduk and Andria are in a lowland (100 meters). Add information  (rows 93 - 95) regarding geography. Where exactly was found a new species in Bulgaria - Starosel (350 meters) or in the mountains (1500 meters). This is also important information.

Author Response

Reviewer 1

Reviewer 2 Report

Please see pdf attached.

Author Response

Reviewer 2
